# Hormonal Interplay Leading to Black Knot Disease Establishment and Progression in Plums

**DOI:** 10.3390/plants12203638

**Published:** 2023-10-21

**Authors:** Ranjeet Shinde, Murali-Mohan Ayyanath, Mukund Shukla, Walid El Kayal, Praveen Saxena, Jayasankar Subramanian

**Affiliations:** 1Department of Plant Agriculture, University of Guelph, Guelph, ON N1G 2W1, Canadaayyanath@uoguelph.ca (M.-M.A.); mshukla@uoguelph.ca (M.S.); psaxena@uoguelph.ca (P.S.); 2Department of Plant Agriculture, University of Guelph, Vineland Station, ON L0R 2E0, Canada; we21@aub.edu.lb; 3Faculty of Agricultural and Food Sciences, American University of Beirut, Beirut 1107-2020, Lebanon

**Keywords:** *Apiosporina morbosa*, phytohormones, plant defense, plant immunity, plant–pathogen interactions

## Abstract

Black Knot (BK) is a deadly disease of European (*Prunus domestics*) and Japanese (*Prunus salicina*) plums caused by the hemibiotrophic fungus *Apiosporina morbosa*. After infection, the appearance of warty black knots indicates a phytohormonal imbalance in infected tissues. Based on this hypothesis, we quantified phytohormones such as indole-3-acetic acid, tryptophan, indoleamines (N-acetylserotonin, serotonin, and melatonin), and cytokinins (zeatin, 6-benzyladenine, and 2-isopentenyladenine) in temporally collected tissues of susceptible and resistant genotypes belonging to European and Japanese plums during of BK progression. The results suggested auxin-cytokinins interplay driven by *A. morbosa* appears to be vital in disease progression by hampering the plant defense system. Taken together, our results indicate the possibility of using the phytohormone profile as a biomarker for BK resistance in plums.

## 1. Introduction

The fungus *Apiosporina morbosa* causes a tumor-like fungal disease called ‘Black knot’ (BK) in both of the commercially important plum species, i.e., European (*Prunus domestica*) and Japanese (*Prunus salicina*) plums [1]. The disease is identified by the appearance of warty, black knots that can range in size, from half an inch to over a foot in length [1]. Once established, the disease progresses, and the condition worsens. On occasions, old knots are covered by a powdery pink or white saprophytic fungal growth and are often attacked by other insects, which further damages the trees. Multiple infections cause trees to lose vigor, bloom poorly, become increasingly unproductive and pre-dispose them to further damage, such as susceptibility to winter injury [2]. BK causes severe economic losses in North America [3,4]. What is more, BK is an understudied disease, which could be due to the difficulty of testing in controlled conditions, unlike other fungal diseases.

Plant-invading pathogens based on nutrient acquisition are classified as necrotrophs, biotrophs, and hemibiotrophs [5]. Hemibiotrophs show a biotrophic phase of lifestyle in the beginning and later enter the necrotrophic phase of lifestyle, hence *A. morbosa* was considered to be a hemibiotrophic fungus because of its biotrophic behavior at the early stage of infection and then conversion into necrotrophic phage at the later stage [6]. The defense network of plants is very complex, and it has been shown to be manipulated by phytohormones such as auxins, cytokinins, and others [7]. Moreover, different strategies are used by phytopathogens to manipulate the complex plant defense system to enter and colonize the plant, thus establishing the disease [8]. A common strategy that pathogens use is the manipulation of hormone signaling to promote disease [9]. Phytopathogens exploit the plant’s defense network by either directly producing, making local plant cells at the point of infection to produce, or manipulating the signaling pathway(s) of the plant’s defense system-related hormones. Many phytopathogens produce hormones that contribute to virulence and thus hormones may be considered virulence factors [10].

The knot-forming nature of the BK disease points to the possible involvement of phytohormones like auxin and/or cytokinins in symptom development. Pathogen-induced hormones such as auxins probably could enhance the plant’s susceptibility to hemibiotrophic pathogens by directly or indirectly impacting the plant’s defense-related hormones [7,11]. Based on the nature of the disease and symptom development we hypothesize that phytohormones, specifically auxin and/or cytokinins are involved in the disease establishment and progression. Thus, the primary objective of this study is to underpin phytohormone titer change during knot progression in both types of commercial plums. Understanding the changes in phytohormone titers after BK disease infection and progression in plums might help in developing a phytohormone-based biomarker for the future breeding of plums. 

## 2. Results

BK is a very slow disease taking almost 2 years for a full manifestation of the knots. Due to its extremely slow nature and environment-dependent infection process, consistent controlled infection is futile. Thus, one must rely on the natural infection of susceptible genotypes in a genetically diverse population over an extended period of at least 7–8 years before a phenotypic judgment on resistant genotypes can be passed. This perhaps is the single most important reason for this disease being minimally researched. Based on multiple years of observations, we developed a scoring system for disease progression depicting five clear stages of black knot development in plums [1]. Briefly, the disease initially starts as a small swelling on the wood in early spring, which bulges as the season progresses. The bark ruptures and the fungus grows into a velvety green mat on the swelling. During late summer and early fall, the green velvet is replaced by a black, cankerous layer, hence the name black knot. For the current study, samples from different stages were collected as described previously, and concomitant branches from resistant genotypes were analyzed for changes in phytohormone titers, to see if phytohormones can be a maker to identify resistance in plums. Due to the long duration of the infection and setting of disease, the samples from resistant varieties should be used as a baseline value, as even a normal-looking healthy stem in a susceptible variety still harbours fungal spores and, thus, internal infection [1]. The presence of such internal infection is bound to corrupt the titer values and hence taking readings on healthy branches of susceptible varieties was not performed. 

Susceptible plum genotypes (European and Japanese) were analyzed for changes in phytohormones titers during the progression of BK disease in comparison with unaffected/resistant genotypes. Phytohormones titers were affected by BK infection in susceptible plum genotypes irrespective of their origin. In general, TRP (Figure 1A,B), BA (Figure 2E,F), and NAS (Figure 1E,F) titers were higher in the European plum as compared to the Japanese plum. For instance, TRP ranged from 4000 to 18,000 ng/g in European plum but only from 1000 to 6000 ng/g in Japanese plum. SER was significantly higher in the later stages (Stages 3–5) in the Japanese plum as compared to the European plum (Figure 1C,D). TRP (Figure 1A,B), BA (Figure 2E,F), and GA (Figure 2G,H) were approximately 2-fold lower in susceptible genotypes of both types of plums. However, an approximately 2-fold increase was noted in the titers of IAA (Figure 1G,H), zeatin (Figure 2A,B), ip (Figure 2C,D), and NAS (Figure 1E,F). 

### 2.1. Effect of BK Disease Progression on TRP and Indoleamines Levels

In general, resistant European plums had higher levels of TRP than the rest (Figure 1A,B). At stage 5, European plums had almost 20,000 ng/g of TRP against less than 10,000 ng/g in Japanese plums. Overall TRP was low in Japanese plums throughout the BK development (Figure 1B). However, in both plums, TRP levels peaked in resistant genotypes, during the period corresponding to the late stage of BK incidence (Figure 1A,B), when the fungus had probably stopped growing. This corresponds to late fall when the trees will probably be preparing for winter [12].

NAS peaked in stage 3 of BK development and then dropped rapidly in both plum varieties (Figure 1E,F). Interestingly NAS was higher in the BK during stages 1 and 3 (850–980 ng/g, respectively), but otherwise, it was pretty close between susceptible and resistant genotypes in both plums, barring an anomalous stage 2 in the European plum (Figure 1E,F). SER was very high during the equivalent of stages 2 and 3 in the resistance genotypes and then dropped steeply (Figure 1C,D). In European plums, SER was below the detectable level in the other three stages (Figure 1C). However, in the resistant Japanese plums it was significantly present in the later stages as well (Figure 1D). This suggests that the elevated levels of SER might confer resistance to BK in plums. However, the counterpart of SER, melatonin (MEL), was not present in detectable levels in all.

### 2.2. Effect of BK Disease Progression on Auxin Levels

Since IAA is a natural auxin, we measured IAA levels during BK development. Auxin levels were significantly higher in European plum knots in all stages of disease progression than in Japanese plum knots (Figure 1G,H). IAA titers in resistant genotypes were almost negligible and, during the 4th and 5th stages, it was not detectable or barely there in both plums (Figure 1G,H). In both plums, IAA levels peaked until stage 2/3 (9000 to 13,500 ng/g) and then started to decline (Figure 1G,H). This suggests that IAA is probably produced by the fungus as the BK symptoms peak. One of the well-known manifestations of auxin is promotion of cell division. Perhaps the high levels of auxin, beyond the normal levels in plum create an atmosphere of uncontrolled cell division at the infected sites resulting in the gnarly black knots. Together both TRP and auxin data led us to conclude that the TRP is converted by the fungus into IAA as the fungus establishes BK disease, while TRP remains intact in resistant genotypes.

### 2.3. Effect of BK Disease Progression on Cytokinins Levels 

Both zeatin (Figure 2A,B) and ip (Figure 2C,D) were higher in the susceptible genotypes than the resistant genotypes, while the contrary was noted for BA (Figure 2E,F). Zeatin (Figure 2A,B) and ip (Figure 2C,D) followed a similar trend to the other hormones, steadily increasing until stage 3 from 200 to 420 ng/µl and then tapering off to a low at stage 5. Zeatin levels were significantly higher in Japanese plums than in European plums until stage 3 in the susceptible genotypes (Figure 2A,B). However, ip was similar in both types of plums (Figure 2C,D). Overall, zeatin (Figure 2A,B) content was much higher, reaching ~800 ng/g tissue, while ip (Figure 2C,D) reached a maximum of only 25 ng/g. 

### 2.4. Effect of BK Disease Progression on GA Levels

Significantly high levels of GA were observed in all five of the stage equivalents in the resistant genotypes of both plums (Figure 2G,H). Resistant European plums had a much higher level (2–3-fold) of GA than Japanese plums (Figure 2G,H). On the knots, it was significantly low in all stages and there was no discernable trend in GA levels in the susceptible genotypes (Figure 2G,H). Like other hormones, GA levels also maxed during the third stage equivalent in the resistant genotypes (Figure 2G,H). This suggests that the GA is all endogenous and the fungus perhaps utilizes the endogenous GA as seen by the lower values. The cells (mostly the wood) in the BK-infected areas also have very elongated cells, which is often a manifestation of GA (Figure 3). 

### 2.5. PCA on Phytohormones during BK Disease Progression

Correlation among the phytohormone levels was determined using PCA of the susceptible genotypes of European plum, where approximately 50% variability was explained by PCA I and approximately 40% by PCA II (Figure 4). However, in the resistant genotypes, approximately 45–65% of the variability was explained by PCAI and 30% by PCA II. In the PCA of susceptible genotypes of Japanese plum, approximately 80% of the variability was explained by PCA I and approximately 10% by PCA II. However, in the resistant genotypes, approximately 70% of the variability was explained by PCAI and 25% by PCA II. At stage 1, in European susceptible genotypes except for ip, other compounds were positively correlated. IAA and NAS appeared to be strongly correlated (>0.8). IAA was strongly negatively correlated with ip (>0.9). At stage 5, TRP, IAA, NAS, and GA appeared to be highly correlated (>0.6). IAA and ip were moderately negatively correlated (>0.4–0.6). Interestingly, this was not the case in resistant genotypes of the European plum. Here, in stage 5, all analyzed compounds except SER were positively highly correlated (>0.7). Intriguingly, at stage 5 of resistant European genotypes, TRP, SER, and IAA were unrelated to other compounds. NAS was highly correlated to BA and ip but not to zeatin. Also, GA, NAS, BA, and ip were noted highly correlated (>0.6) (Figure 4). These results are in agreement with the phytohormonal responses at specific stages (Figure 1 and Figure 2). 

In Japanese susceptible plum genotypes, except for SER and BA, all other analyzed compounds were positively correlated (Figure 4). TRP, NAS, zeatin, ip, and GA were noted to be strongly positively correlated (>0.9). Interestingly, a similar trend was noted at stage 5 except NAS being negatively correlated with other analyzed compounds (>0.8). In the resistant Japanese genotypes, all compounds except BA and SER were highly positively correlated (>0.7). The moderate (0.4–0.6) negative correlation between BA and other analyzed compounds is noteworthy. Surprisingly, at stage 5, except IAA, all of the compounds were positively correlated, whereas BA was exceptionally poorly correlated to other compounds (<0.3). At stage 5, uniquely, SER was noted to be strongly correlated to TRP and NAS (>0.7). This is noteworthy of undetected and unrelated IAA in both the resistant genotypes at stage 5 (Figure 4). 

Overall, our results suggest that higher endogenous levels of TRP, GA, and SER contribute to BK resistance while auxin and cytokinins promote susceptibility. In particular, higher levels of TRP seem to be a strong indicator of BK resistance. However, it must also be noted that the extremely high levels of IAA produced by the knots are suggestive of this auxin being produced by the fungus, rather than fungus–host interaction. Such high levels of auxin and cytokinins, probably produced by the fungus along with the utilization of gibberellic acid from the host plants results in uncontrolled cell division at the infected sites leading to black knots. Although both auxin and cytokinins (zeatin and ip) are higher in susceptible genotypes, the levels of cytokinins are only a fraction of IAA, indicating a possible antagonism than synergism between the two hormones.

## 3. Discussion

Phytohormones perturbance is certainly triggered by *A. morbosa*, during BK disease progression. Overall, the susceptible genotypes exhibit significantly higher levels of auxin and cytokinins and lower levels of indolamines than the resistant genotypes. This opens up the question of identifying the possible mechanisms of how plums counter a fungal infection through phytohormones and whether the contents of phytohormones can be used as biomarkers. Pathogen-induced hormones such as auxin probably could enhance the plant’s susceptibility by impacting plant defense [7,9,13].

In general, European plums had significantly higher phytohormone titers, which could be related to the hexaploidy nature of the European plum [14]. As a hexaploid, the European plum could possess multiple copies of related genes, thus resulting in higher titers. However, not all the genes become overexpressed in the polyploid species as compared to its diploid species [15]. This could explain the lower titers of ip and SER in European plum. Conversely, the hexaploidy in European plums is more due to the introgression of genes from *P. spinosa* and *P. cerasifera* [16], which could explain the differences as well. 

In resistant plums, the TRP is stored as a part of dormancy acquisition [12], while they are lost in susceptible plums due to their conversion to IAA. It is well known that auxin synthesis by phytopathogenic fungi using the host plant’s TRP as in Arabidopsis [17,18]. Based on these observations, it is tempting to speculate that the fungus is facilitating such a diversion of indoleamines to IAA, which is further fueled by circumstantial evidence [19,20].

### 3.1. Effect of Auxins on BK Disease Progression

The most common form of auxin in plants is indole-3-acetic acid (IAA). The involvement of auxins in plant defense response is a subject of recent interest and, for a long time, auxins were not considered a primary line of plant defense [9]. The role of auxin signaling is dependent on the nature of the pathogen and the type of host–pathogen interaction. Existing research points to the fact that increased auxin signaling enhances disease resistance against necrotrophic pathogens while promoting disease in plants infected with biotrophic and hemibiotrophic pathogens [7,13]. For instance, increased IAA levels have been observed after infection by necrotrophic fungi like *Botrytis*, *Rhizoctonia*, and *Alternaria* in different plants like arabidopsis, rice, and lupin [21,22,23,24]. Further, exogenous application of auxins increased the disease susceptibility in biotrophic and hemibiotrophic pathogens such as *Phytophthora parasitica* in arabidopsis [25], and *Magnaporthe grisea*, *Pseudomonas syringae*, and *Xanthomonas oryzae* in rice [26]. 

IAA promotes disease in many plant–pathogen interactions by acting as a plant hormone and as a trigger of the microbial signal. As a plant hormone, IAA increases host susceptibility by modulating host signaling and physiology, and as a microbial signal, IAA directly boosts pathogen virulence [9]. To take over the host, pathogens use different strategies including the production of auxins, interrupting auxin signaling in which auxin contributes to resistance and boosts auxin signaling, in which auxin increases susceptibility [7]. Moreover, boosted auxin signaling can increase disease symptoms, development of galls/knots and feeding sites, and/or suppress other defense responses [27].

Gall/knot-forming phytopathogens like *Agrobacterium tumefaciens* and parasitic nematodes interfere with host auxin physiology to encourage hypertrophy and hyperplasia which leads to gall/knot formation. Further, pathogens can either synthesize auxin themselves [28,29], stimulate local plant cells at the site of infection to synthesize auxin [30], or divert auxin flow in the plant towards the site of infection to stimulate hypertrophy and hyperplasia [31]. However, which of these mechanisms is being used by *A. morbosa* will be a subject of interest for future research. Knots reach their maximum size during the 3rd developmental stage (Figure 1a). Similarly, IAA levels also increased steadily until the 3rd stage, suggesting the definite contribution of the fungus to auxin production. At the 4th and 5th stages, the knots stop growing but start ‘maturing’ and proceed towards reproductive stages [32]. This could explain the lower levels of IAA in these stages (Figure 1G,H). However, during the 5th stage in resistant genotypes, the TRP (Figure 1A,B) titers were at their peak, but IAA (Figure 1G,H) was below the detectable level, possibly indicating the preparation of resistant genotypes for the winter while, in susceptible genotypes, TRP was extremely low at the 5th stage (Figure 1A,B). It is very likely that much of the TRP at this stage has been converted to IAA by the fungus. Thus, susceptible genotypes continue to be under BK stress and could not prepare well for the winter as compared to the resistant genotypes. It is well known that winter injury is augmented in BK-affected trees and such augmentation of winter injury could be related to inadequate preparation of susceptible trees [2].

### 3.2. Effect of Cytokinins on BK Progression

Zeatin and ip are the major plant cytokinins, and ip is a precursor in zeatin synthesis [33]. Our results suggest that zeatin is the major cytokinin while ip and BA are present in comparatively minor quantities in plums. Cytokinins play a major role in cell division, shoot development, and delay senescence [34,35]. Further, cytokinins also have a role in plant immunity [36,37]. The involvement of cytokinins in tumor, knot, and gall-forming diseases is well known [38]. An early wave of cytokinins can elevate host susceptibility but the precise phenomenon is unknown [39]. Interestingly, low levels of cytokinins elevate disease susceptibility while higher levels tend to improve resistance [40,41], although the mechanism of this is still unknown [42]. It is proposed that cytokinin production and/or signaling might be advantageous to biotrophic and hemibiotrophic phytopathogens for successful disease establishment [39,42].

Phytopathogens often synthesize cytokinins [39,43,44,45] and can enhance the plant’s signaling/sensing of cytokinin [46,47]. For example, several biotrophic and hemibiotrophic pathogens produce cytokinins to redirect plant organogenesis to the site of infection, leading to symptom development like hyperplasia or gall/tumor formation [11,39,48,49].

It is proposed that cytokinins produced by biotrophic and hemibiotrophic fungi might be essential to contributing to the disease and symptom development [50]. Knocking out the cytokinin biosynthetic gene cks-1 has been shown to elevate resistance in rice against *Magnaporthe oryzeae*, a hemibiotrophic fungus [11], while exogenously applied cytokinin increased the susceptibility of *Nicotiana benthamiana* to *Pseudomonas syringae* pv. tomato [51]. *A. morbosa* might be producing zeatin and ip for the fungal establishment and disease progression in susceptible genotypes after infection. Probably, ip might be used by *A. morbosa* as a precursor in zeatin synthesis, as the trends of both cytokinins show similarity in susceptible genotypes. 

### 3.3. Role of Indoleamines on BK Disease Progression 

TRP titers accumulated in the resistant genotypes of the European and Japanese plums might be related to winter-readiness and the next season [12]. Lower TRP titers in susceptible genotypes during fall suggest resource allocation, while countering biotic (fungal) stress. Further, in susceptible genotypes, TRP might be converted to IAA by *A. morbosa*, thus resulting in lower titers. In addition, SER and MEL are shown to play a role in alleviating biotic stresses in plants [20]. The conversion of TRP to IAA might be the strategy adopted by *A. morbosa* to divert the TRP from the indoleamine pathway, probably to increase the susceptibility of the tree to the disease, thus helping the fungus to establish further in the tree.

Higher titers of SER in resistant genotypes of plums could impart resistance against BK. SER inhibited the sporulation of *Stagonospora nodorum* by preventing spore formation and maturation within pycnidial structures in wheat [52]. Moreover, exogenously applied SER induced defense gene expression and cell death in rice suspension cultures, and increased resistance against rice blast infection in plants [53]. In addition, SER was embedded in cell walls, reinforcing the cell wall strength, a key first line of defense against fungal infection as demonstrated in rice [54]. Extremely low levels of SER, coupled with high levels of NAS, in susceptible genotypes suggest the conversion of SER to NAS in susceptible genotypes, although this requires further research. Higher levels of NAS also suggest that NAS could not lead to sustained MEL synthesis. It is possible that the synthesized MEL either became conjugated to 2-hydroxy melatonin or degraded immediately after synthesis or both. 

In conclusion, our results suggest that the fungus *A. morbosa*, is involved in the production of phytohormones during the process of disease establishment. Further, *A. morbosa* deprives the plant of TRP as the infection progresses, resulting in susceptibility and predisposing the trees to winter injury. Lack of TRP in combination with increased levels of auxin and cytokinins at various stages promotes disease susceptibility, which aids in disease establishment and progression. It is natural at this juncture to question whether JA and/or SA have a role to play in this interaction. Our preliminary analyses have revealed rather interesting behaviour of JA and SA in relation to BK infection in plums [55], which warrants separate in-depth analyses and discussion and, hence, those results will be published separately later. Altogether, our results show that phytohormones have a role in BK resistance/susceptibility and perhaps a phytohormone profile can be used as a marker to breed for BK resistance. 

## 4. Materials and Methods

### 4.1. Sample Collection

From the University of Guelph’s plum breeding program, 15-year-old European (*Prunus domestica*) and Japanese (*Prunus salicina*) plum trees were selected for the current study. Phytohormonal analysis was performed using genotypes ‘Vision’ and ‘Veeblue’ from the European plum and ‘Vampire’ and ‘Shiro’ from the Japanese plum, which are extremely susceptible to the black knot disease, and resistant genotypes V982014 and V911415 from the European plum and ‘Underwood’ and ‘Redcoat’ from the Japanese plum. The resistance and susceptibility of these genotypes were established based on the assessment of BK incidence and progression over 10 years, with limited disease control in a diverse population consisting of ~150 genotypes of each Japanese and European plum [55]. Samples from black knots at five different developmental stages (based on the stage of the infection, from the end of May until mid-October at approximately 3-week intervals) were collected from susceptible genotypes and stored at −80 °C after flash freezing with liquid nitrogen. In the case of resistant genotypes, at the same time points mentioned above, the branches of comparable age and size were used for sample collection.

### 4.2. Freeze Drying and Grinding

Freeze drying of all the woody samples was carried out for 48 h with a FreeZone 4.5L −50 °C Benchtop Freeze Dryer (Labconco Corporation, 8811 Prospect Avenue, Kansas City, MO 64132, USA) immediately after sample collection from the field. Then, the freeze-dried samples were stored at −80 °C until they were ground using an IKA^®^ A 11 basic Analytical mill (IKA Works, Inc., 2635, Northchase Parkway SE, Wilmington, NC, USA) with liquid nitrogen.

### 4.3. Hormone Extraction, Identification, and Quantification

All the phytohormones from ground woody samples were extracted using the methanol double extraction method. Briefly, 100 mg of freeze-dried, powdered woody samples were extracted with a solvent mixture (methanol: formic acid: milli-Q water = 15:1:4). Samples were then held at −20 °C for 30 min and spun down (15 min, 4 °C, 14,000 rpm) and the supernatant was removed. A second extraction was performed on the same sample using similar conditions to those described above and the supernatants were pooled. Solid phase extraction (Oasis^®^ HLB 1cc (30 mg), Waters Canada, Mississauga, ON, Canada) was deployed to concentrate the samples before eluting them in 200 μL methanol. Later, the eluant was filtered through a 0.22 μM centrifuge filter (Millipore, Burlington, MA, USA; 1 min, 13,000 rpm). All standards were analytical grade and purchased from Sigma Aldrich, Oakville, ON, Canada. Phytohormones were separated by reverse-phase liquid chromatography (ultra-performance liquid chromatography system (UPLC); LC-40D XS, Shimadzu, Tokyo, Japan) with an injection of a 5 μL aliquot of sample onto the Shim-pack Scepter LC column (2.1 × 50 mm, 1.9 μm; Mandel Scientific Company, Guelph, ON, Canada). Metabolites were separated with a gradient of solvents A (0.1% formic acid) and B (100% methanol) with initial conditions at 95% A (5% B) increased to 5% A (95% B) over 4 min using a curve of 0. The column temperature was 40 °C and the flow rate was 0.2 mL/min. Metabolite peaks were identified by comparison to standards and quantified by a standard curve generated using a similar separation method and gradient conditions. Phytohormones were detected using a single-quadrupole mass spectrometer (LCMS 2020, Shimadzu, Japan) in single ion recording mode (SIR). The majority of the compounds were detected in positive mode with a cone voltage of 10 for mass to charge (*m*/*z*) of 205 (tryptophan (TRP)), 177 (serotonin (SER)), 219 (N-acetylserotonin (NAS)), 176 (indole-3-aceic acid (IAA)), 204 (2-isopentenyladenine (ip)), 220 (zeatin), 226 (6-benzylaminopurine (BA)), and in negative mode with a cone voltage of 10 for mass to charge (*m*/*z*) of 345 (gibberellic acid (GA)). In all cases, the probe temperature was set to 250 °C with a gain of 5; capillary voltage (positive and negative) was set to 0.5 kV. The linear range for all eight compounds was 1.53 ng/mL–6.25 μg/mL. 

### 4.4. Statistical Analysis

The study was performed using four biological replicates from each genotype and two genotypes from each group, thus eight technical replicates from each group. Hormonal data of the European and Japanese plums were analyzed together using general linear mixed models (proc GLIMMIX) in SAS v9.4 (SAS Institute Inc., 111, Hampton Woods Ln, Raleigh, NC 27607, USA). Shapiro–Wilk normality tests and studentized residual plots were used to test error assumptions of variance analysis including random, homogenous, and normal distributions of error. Outliers were removed using Lund’s test. Means were calculated using the LSMEANS statement, and significant differences between the treatments were determined using a post hoc LSD test α ≤ 0.05 and are mentioned in each figure.

The phytohormonal contribution in BK disease development was analyzed using principal component analysis (PCA) at the 1st and 5th stages. Here, PCA was conducted separately on phytohormonal data from resistant and susceptible genotypes of European and Japanese plums, using PROC PRINCOMP in SAS v9.4 (SAS Institute Inc., 111, Hampton Woods Ln, Raleigh, NC 27607, USA).

## Figures and Tables

**Figure 1 plants-12-03638-f001:**
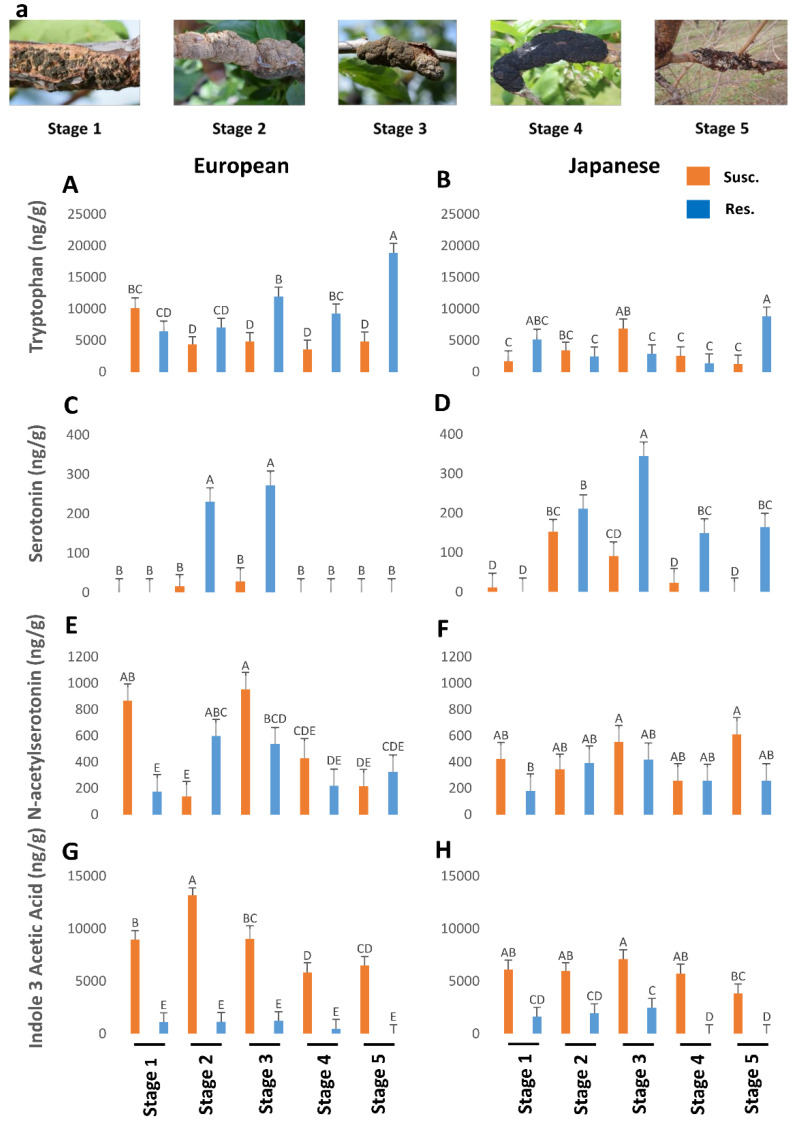
The titers of tryptophan (*p* < 0.05) (**A**,**B**), serotonin (*p* < 0.05) (**C**,**D**), N-acetylserotonin (*p* < 0.05) (**E**,**F**), and indole-3-acetic acid (*p* < 0.05) (**G**,**H**) in genotypes of European and Japanese plums resistant and susceptible to Black Knot (BK) disease at five different BK developmental stages (1–5) (**a**). Stage 1 is the appearance of visual symptoms of BK while stage 5 is the most developed knot. Different letters denote statistical significance and error bars represent means ± SEM (ng/g DW) for all phytohormones. Interaction *p*-values were reported.

**Figure 2 plants-12-03638-f002:**
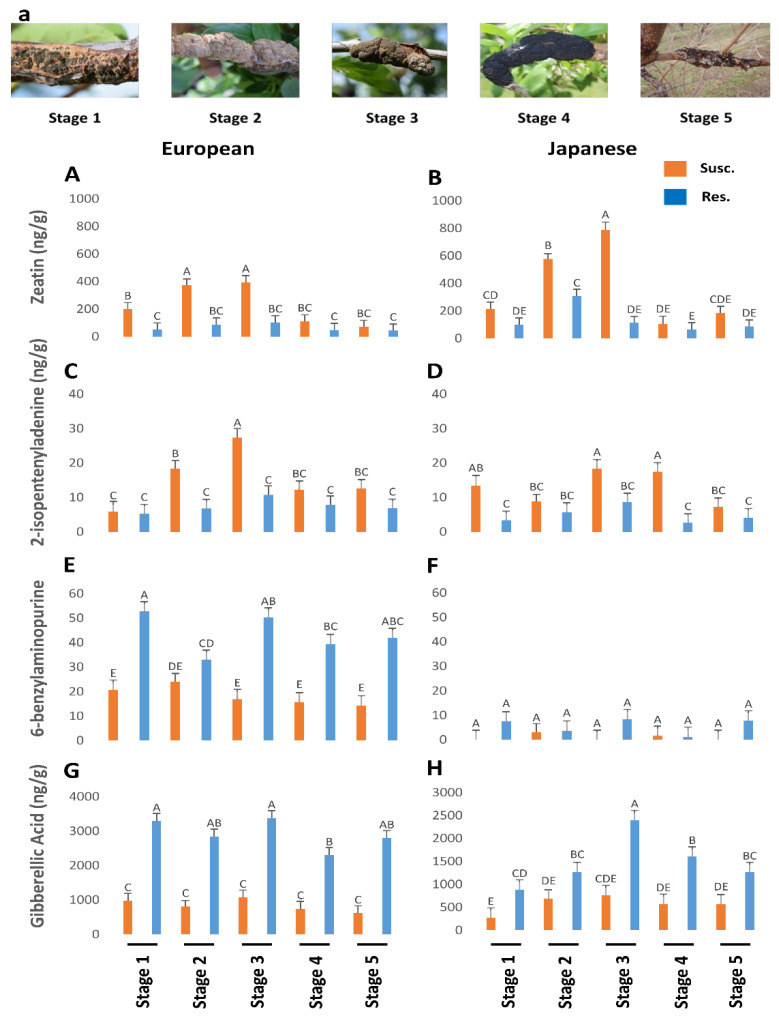
The titers of zeatin (*p* < 0.05) (**A**,**B**), 2-isopentenyladenine (*p* < 0.05) (**C**,**D**), 6-benzylaminopurine (*p* = 0.47) (**E**,**F**), and gibberellic acid (*p* = 0.19) (**G**,**H**) in genotypes of European and Japanese plums resistant and susceptible to Black Knot (BK) disease at five different BK developmental stages (1–5) (**a**)). Stage 1 is the appearance of visual symptoms of BK while stage 5 is the most developed knot. Different letters denote statistical significance and error bars represent means ± SEM (ng/g DW) for all the phytohormones. Interaction *p*-values were reported.

**Figure 3 plants-12-03638-f003:**
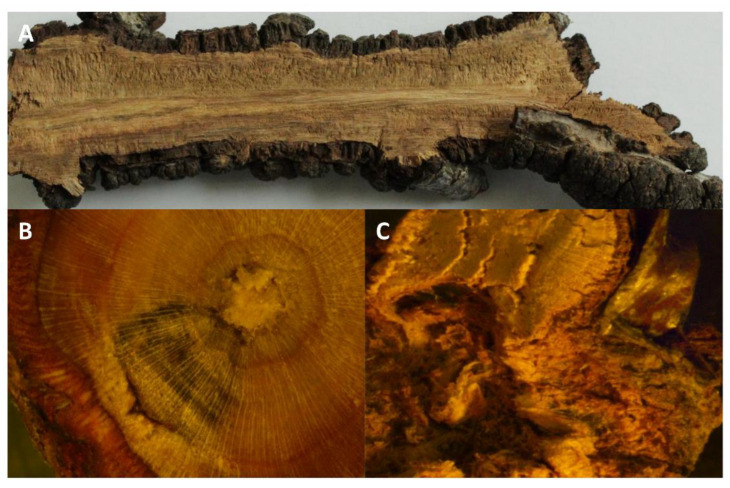
Longitudinal (**A**) and cross-sections (**C**) of matured Black Knot (BK) and the cross-section of a branch from the resistant genotype (**B**) unfolded the internal elongated cells representing the manifestation of gibberellic acid (GA), auxin, and cytokinin imbalance.

**Figure 4 plants-12-03638-f004:**
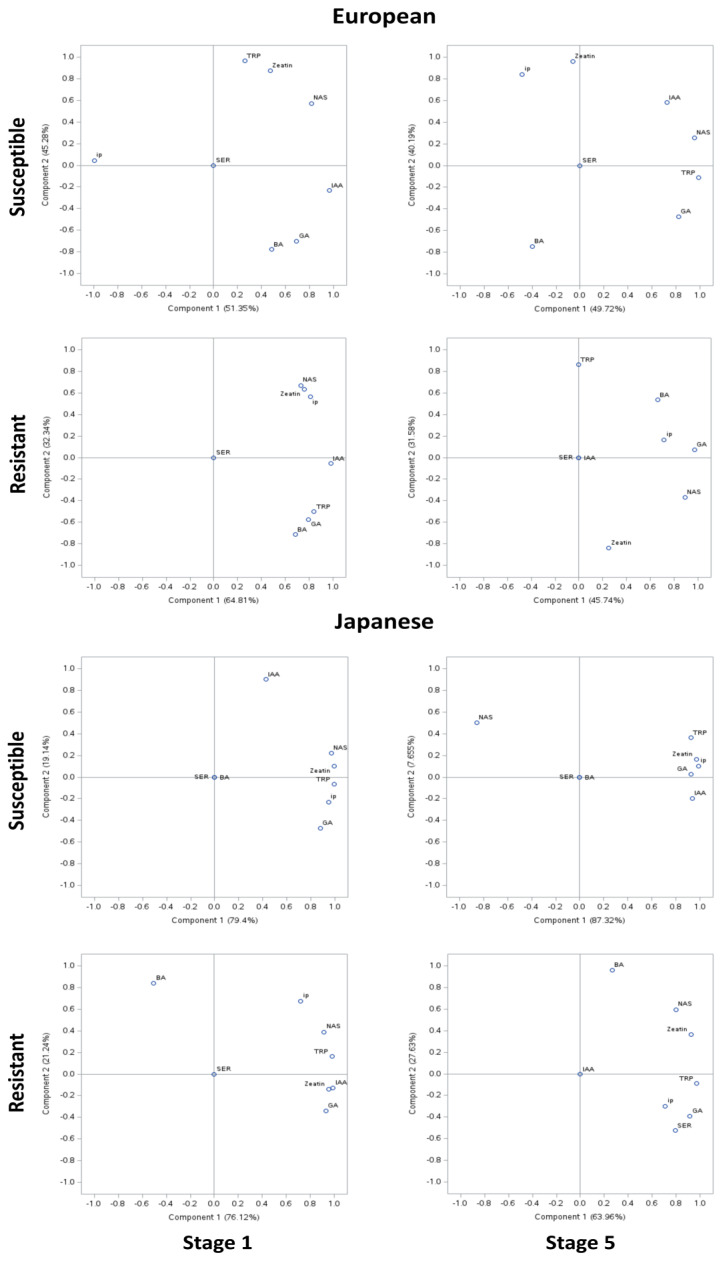
Principal component analysis (PCA) of phytohormonal contents of the Black Knot (BK) disease-resistant and susceptible genotypes of European and Japanese plums at the extreme ends of the BK progression, i.e., the 1st (beginning of the infection) and the 5th (highly developed knot) stages. Phytohormones tryptophan (TRP), N-acetylserotonin (NAS), serotonin (SER), indole-3-acetic acid (IAA), zeatin, 2-isopentenyladenine (ip), 6-benzylaminopurine (BA), and gibberellic acid (GA) are labeled at data points. The distance of the data point from the center is directly proportional to the variability explained by PCA I and II. The angle of the line passing through the center connecting two data points is inversely proportional to the correlation between the phytohormones.

## Data Availability

Data sharing is not applicable to this article.

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
