# Peer review of "Hormonal Interplay Leading to Black Knot Disease Establishment and Progression in Plums"

_plants, 2023, doi:10.3390/plants12203638_

Round 1
Reviewer 1 Report
The manuscript by Shinde et al. investigates the phytohormone profile during the progression of Black Knot (BK) disease in plums. The authors aim to gain insights of disease development and potentially use hormone profiling in the future to identify resistant plum phonotypes. The authors indeed found significant differences in phytohormones between resistant and susceptible phenotypes indicating initial success. The authors' mechanistic interpretation of the data is good and insightful. However, some important controls and data are missing, and the statistical analysis methods are not explained well and are uncommon. These issues should be addressed before the manuscript is acceptable for publication.
Statistical analysis:
1) Figs. 1 and 2 contain numerous statistical tests; however the test design is strange and information is missing. What is the sample number (n) for the test? Is it 8 as explained in methods, i.e., 4 samples pooled from each resistant/susceptible genotype to give 8 total? The authors mentioned removing outliers, how many were removed for each sample and how exactly? Since all these measurements were taken from the same sample, then it would be incorrect to remove, for example, zeatin datapoint from sample 1 for European resistant plum from stage 1 but keep GA datapoint from the same sample as these would be tied together.
2) The authors provide P value in the legend of Fig. 2. What is the P value from, One way ANOVA? What is it comparing? How can both panels have the same P-value (e.g., Fig. 2 E and F, P=0.47) when one seems to have significant differences (2E) and the other does not (2F). Were both European and Japanese plums pooled together for analysis? This would be a wrong approach as they might have different baseline hormone levels (e.g., TRP).
3) From the methods, it seems the authors have used Fisher’s LSD, which does not correct for multiple comparisons. Why not use Tukey’s or Bonferroni? This analysis method seems strange.
4) How exactly was the PCA analysis done? Which dimension was reduced, was it the replicates? Why is SER always at the center of the figure, was it used for a normalization? Where is the data for Japanese plums? It’s also unclear how the correlation values were obtained from the PCA plot, a visual representation on the graph would be helpful. I am familiar with the use of PCA for analyzing large-scale transcriptomic data, however this method of looking for correlations, to me, seems uncommon. A simpler and more straightforward correlation analysis might be better, or the authors should make it more clear why PCA was used for this analysis.
5) If anything, a PCA of reducing the dimension across all hormones (and, possibly, all stages of infection) and seeing if Susceptible and Resistant genotypes cluster together no matter if, for example, ‘Vision’, ‘Veeblue’, ‘Vampire’ or ‘Shiro’ was used for susceptible genotype is more relevant if authors aim to use hormone profiling as a marker of resistance. This would also give justification for simply grouping ‘Vision’ and ‘Veeblue’, for example, together as the authors have done (as far as I can tell from methods).
Controls:
1) The authors have not provided phytohormone levels from uninfected trees. This seems strange as this would have been a natural baseline for comparing hormone levels throughout disease progression, especially for tryptophan, for example, that shows strong differences between Japanese and European plums.
2) The authors have not measured the defense hormone levels (SA and JA). They do mention it in the discussion as a future work, however, it might be important to measure them now. Especially SA, which, to my knowledge, is relatively easy to measure and which is the primary hormone governing the plant-pathogen interaction for the biotrophic pathogens (at least in Arabidopsis). It is possible that the differences between these phenotypes might be better explained by consistently higher SA levels across all stages in resistant phenotypes compared to susceptible ones. The authors should either measure it, or provide justification of why it is not feasible to do so now.
Author Response
Comments and Suggestions for Authors
The manuscript by Shinde et al. investigates the phytohormone profile during the progression of Black Knot (BK) disease in plums. The authors aim to gain insights of disease development and potentially use hormone profiling in the future to identify resistant plum phonotypes. The authors indeed found significant differences in phytohormones between resistant and susceptible phenotypes indicating initial success. The authors' mechanistic interpretation of the data is good and insightful. However, some important controls and data are missing, and the statistical analysis methods are not explained well and are uncommon. These issues should be addressed before the manuscript is acceptable for publication.
Statistical analysis:
1) Figs. 1 and 2 contain numerous statistical tests; however the test design is strange and information is missing. What is the sample number (n) for the test? Is it 8 as explained in methods, i.e., 4 samples pooled from each resistant/susceptible genotype to give 8 total? The authors mentioned removing outliers, how many were removed for each sample and how exactly? Since all these measurements were taken from the same sample, then it would be incorrect to remove, for example, zeatin datapoint from sample 1 for European resistant plum from stage 1 but keep GA datapoint from the same sample as these would be tied together.
Yes, there were 8 samples (4 trees x 2 genotypes) from susceptible and tolerant. When outliers were detected for that tree (student residuals > ± 3.3) the analysis was carried out by removing the outlier as per standard statistical procedures.
2) The authors provide P value in the legend of Fig. 2. What is the P value from, One way ANOVA? What is it comparing? How can both panels have the same P-value (e.g., Fig. 2 E and F, P=0.47) when one seems to have significant differences (2E) and the other does not (2F). Were both European and Japanese plums pooled together for analysis? This would be a wrong approach as they might have different baseline hormone levels (e.g., TRP).
As mentioned in the text, proc glimmix was used to analyze each compound separately. Trends (statistically insignificant) noted in that compound were important to explain the results and hence p-value for that compound was reported to avoid confusion. European and Japanese plums were not analyzed together as their ploidy is different. For ease of understanding all the data were presented in a pooled graphical format.
3) From the methods, it seems the authors have used Fisher’s LSD, which does not correct for multiple comparisons. Why not use Tukey’s or Bonferroni? This analysis method seems strange.
We completely agree that Tukey’s or Bonferroni’s mean separation is better than LSD. However, these samples were field collected and were based on observations. The trees were scored based on a scale developed for the first time for this disease and hence observational error (LSD) was considered, which we think is better ( and confirmed by our statisticians) for such scenario than the Tukey’s or Bonferroni’s mean separation.
4) How exactly was the PCA analysis done? Which dimension was reduced, was it the replicates? Why is SER always at the center of the figure, was it used for a normalization? Where is the data for Japanese plums? It’s also unclear how the correlation values were obtained from the PCA plot, a visual representation on the graph would be helpful. I am familiar with the use of PCA for analyzing large-scale transcriptomic data, however this method of looking for correlations, to me, seems uncommon. A simpler and more straightforward correlation analysis might be better, or the authors should make it more clear why PCA was used for this analysis.
PCA was conducted using proc princomp on each genotype and yes, replicates were reduced. Data presented graphically was overwhelming on a bigger picture and hence PCA was used. Titers of serotonin were below detection limit and hence reported as zero. It was not used for any normalization.
5) If anything, a PCA of reducing the dimension across all hormones (and, possibly, all stages of infection) and seeing if Susceptible and Resistant genotypes cluster together no matter if, for example, ‘Vision’, ‘Veeblue’, ‘Vampire’ or ‘Shiro’ was used for susceptible genotype is more relevant if authors aim to use hormone profiling as a marker of resistance. This would also give justification for simply grouping ‘Vision’ and ‘Veeblue’, for example, together as the authors have done (as far as I can tell from methods).
We understand and appreciate the suggestion but depending on the field relevance these data were needed to explain the results and hence were used.
Controls:
1) The authors have not provided phytohormone levels from uninfected trees. This seems strange as this would have been a natural baseline for comparing hormone levels throughout disease progression, especially for tryptophan, for example, that shows strong differences between Japanese and European plums.
Thank you for pointing this out. However, we need to clarify that the levels of the resistant varieties are used as baseline values here. Ideally one should have a clean tree grown for the same age and one infected with black knot in the same conditions. In reality it is not possible to have such a control as artificial infection of black knot is seldom successful and will take 2-3 years for successful inoculation. As per our earlier research, A. morbosa moves systemically in plum trees (El Kayal et al., 2015, referred to in the ms as [1]). Thus, though the branch does not show visual symptoms and presumed healthy there are spores and ‘internal symptoms’, which will corrupt the results.
We have added a small section in the results (last 2-3 sentences in the first paragraph of the results) to clarify this.
2) The authors have not measured the defense hormone levels (SA and JA). They do mention it in the discussion as a future work, however, it might be important to measure them now. Especially SA, which, to my knowledge, is relatively easy to measure and which is the primary hormone governing the plant-pathogen interaction for the biotrophic pathogens (at least in Arabidopsis). It is possible that the differences between these phenotypes might be better explained by consistently higher SA levels across all stages in resistant phenotypes compared to susceptible ones. The authors should either measure it, or provide justification of why it is not feasible to do so now.
Measuring SA- JA are a very different story and our results are so much that it constitutes a separate manuscript. As the reviewer states measuring SA is easy but measuring JA is not as easy as the others. The behaviour of SA-JA in plums is not the same as in Arabidopsis or in any other herbaceous plant and will warrant extensive discussion and if we add that to this paper the flow and main message will be lost. Hence it will be submitted as a separate manuscript after further confirmation.
Submission Date
13 September 2023
Date of this review
26 Sep 2023 20:55:35

Reviewer 2 Report
Comments and Suggestions for Authors
This paper describes Hormonal Interplay Leading to Black Knot Disease Establishment and Progression in Plums.
Although the work is interesting, there are many concerns that are needed to be addressed.
Ø The abstract part is very poor, not well written, and devoid of results. It should be rewritten in detail describing the problem and the results reached by the authors. In the abstract, the research gap should be improved to strengthen the motivation of the work and considering the important results.
Ø Has the pathogenic fungus been isolated to confirm infection?
Ø What is the severity of the infection? The authors must provide detailed information and results about the pathogen, the symptoms of the infection, and how to confirm the disease.
Ø What are the characteristics of the pathogen that is not defined morphologically or genetically?
Ø There are many grammatical and typographical errors throughout the manuscript. It would be stressful for many readers.
Ø It is necessary to improve the discussion part, particularly in comparison with recent studies.
Ø Conclusion part must contain the importance of paper, the future work and novelty. it is good to include some recommendations.
Ø Please add all abbreviations mentioned in the manuscript to the abbreviation list at the end of the manuscript.
Ø Please check again the references and add the DOI if possible.
Author Response
Comments and Suggestions for Authors
Comments and Suggestions for Authors
This paper describes Hormonal Interplay Leading to Black Knot Disease Establishment and Progression in Plums.
Although the work is interesting, there are many concerns that are needed to be addressed.
Ø The abstract part is very poor, not well written, and devoid of results. It should be rewritten in detail describing the problem and the results reached by the authors. In the abstract, the research gap should be improved to strengthen the motivation of the work and considering the important results.
We have made some minor edits to abstract, but this reviewer is the only person who is finding issue with our language as this has been reviewed by at least 4 outside people with extensive publication record and no one has made anything about the language.
Ø Has the pathogenic fungus been isolated to confirm infection?
We have isolated this pathogen. But we are not sure how that has anything to do with this work on hormones.
Ø What is the severity of the infection? The authors must provide detailed information and results about the pathogen, the symptoms of the infection, and how to confirm the disease.
We have included severity levels in the figures as stage 1-5. It is not clear if the reviewer has seen the black knot disease based on the question, as black knot is not a disease that has any ambiguity. Again, we do not think this has anything to do with the current research.
Ø What are the characteristics of the pathogen that is not defined morphologically or genetically?
The pathogen has been described earlier. First two sentences of the introduction have briefly described the fungus and its general morphology and we have the reference to direct the reader for further description. We are really not sure what this has to do with the current work.
Ø There are many grammatical and typographical errors throughout the manuscript. It would be stressful for many readers.
Okay
Ø It is necessary to improve the discussion part, particularly in comparison with recent studies.
Sorry but this comment is not helpful and very vague.
Ø Conclusion part must contain the importance of paper, the future work and novelty. it is good to include some recommendations.
We have added some recommendations already.
Ø Please add all abbreviations mentioned in the manuscript to the abbreviation list at the end of the manuscript.
Ø Please check again the references and add the DOI if possible.
Thank you but where possible doi have been added. I think the reviewer didn’t notice it as the hyperlinks seem to have been disabled.
Submission Date
13 September 2023
Date of this review
26 Sep 2023 09:16:32
Round 2
Reviewer 1 Report
While the authors have addressed many of the concerns, they still have not addressed some crucial questions.
1) Regrading removing the outliers: the authors still not answer the main question: when say, for example, they detect that the first rep of zeatin is an outlier for stage 1 resistant European plum, do they remove all hormone reading for that sample? While it might be statistically correct to remove an outlier for one hormone only, experimentally it is problematic as the authors look for correlations between hormones. Also, how many outliers were removed?
2) Regarding the P-value: I accept the authors’ arguments for their analysis using LSD, but they still have not answered my question. Line 151: “6-benzylaminopurine (P=0.47) (Fig. 2E,F)” How can both E and F have the same P-value, if one is from European and the other from Japanese plums if they were analyzed separately?
3) Regarding PCA: a) if SER was not used for PCA analysis, it should be omitted from the graph and mentioned as such in the text. b) Fig 4 does not have PCA plots for the Japanese plums, while they are referred to in the text. c) A graphical representation of how PCA plot translates into correlations would be helpful d) Any other examples of how PCA was used before to study correlation?
4) I disagree with the authors' argument about which dimension should be used for PCA. Have the authors tried to do PCA according to resistant/susceptible genotypes at all? If the authors want to show that one can predict whether the plum is resistant or susceptible (as mentioned in the text), then it is literally the purpose of PCA analysis to show that resistant and susceptible genotypes cluster together when the dimensionality of hormone measurements are reduced. Why wasn't this done? The authors mention that "We understand and appreciate the suggestion but depending on the field relevance these data were needed to explain the results and hence were used". I agree that their method can be good for gaining the mechanistic insight, and their explanations make sense, however they cannot use that data to claim that resistant and susceptible genotypes can be distinguished based on hormonal profiles.
5) Regarding SA-JA crosstalk. I know and agree that JA can be notoriously hard to measure. The authors should still provide references in the discussion section that show that SA and JA behave differently in the plums and as such this line of research warrants a separate study.
Author Response
Dear Reviewer
Thank you for the comments again. We have answered all the questions, and I am confident that the manuscript is acceptable for publication now. The results are the first in the world and in a disease that cannot be replicated in controlled conditions and thus will be a good model for perennial crop-pathogen interactions, involving long duration pathogen infection. It also proposes a plausible model/approach to address disease resistance breeding in woody perennials using phytohormones as a biomarker. Both the reviewers agree to that fact which is quite refreshing. While statistics are important to such research, the larger findings should not be lost in the myriad of statistical analyses that are available to us nowadays. Nevertheless, it is an interesting series of comments, and we enjoyed discussing those and answering them. There is no question that these comments have improved the manuscript, and we thank the reviewer for that.
Jay (CA)
-----------------------------
While the authors have addressed many of the concerns, they still have not addressed some crucial questions.
1) Regrading removing the outliers: the authors still not answer the main question: when say, for example, they detect that the first rep of zeatin is an outlier for stage 1 resistant European plum, do they remove all hormone reading for that sample? While it might be statistically correct to remove an outlier for one hormone only, experimentally it is problematic as the authors look for correlations between hormones. Also, how many outliers were removed?
There were three outliers. All the data points were removed for that sample for that stage. We added the outliers back and didn’t notice much difference in the output.
2) Regarding the P-value: I accept the authors’ arguments for their analysis using LSD, but they still have not answered my question. Line 151: “6-benzylaminopurine (P=0.47) (Fig. 2E,F)” How can both E and F have the same P-value, if one is from European and the other from Japanese plums if they were analyzed separately?
For a particular compound the data were analyzed separately and p-value for all way interactions was provided. For that compound, origin, genotype, and tolerance were fixed factors. Except the p-values mentioned in the legend, interactions were statistically significant for remaining compounds.
3) Regarding PCA: a) if SER was not used for PCA analysis, it should be omitted from the graph and mentioned as such in the text. b) Fig 4 does not have PCA plots for the Japanese plums, while they are referred to in the text. c) A graphical representation of how PCA plot translates into correlations would be helpful d) Any other examples of how PCA was used before to study correlation?
SER was used because we need all hormones to understand the interplay. SER can’t be zero. It was labeled as zero only when the peaks were below detection limit of the instrument. Figure 4 has both European (above 4 panels) and Japanese (below four panels) genotypes.
4) I disagree with the authors' argument about which dimension should be used for PCA. Have the authors tried to do PCA according to resistant/susceptible genotypes at all? If the authors want to show that one can predict whether the plum is resistant or susceptible (as mentioned in the text), then it is literally the purpose of PCA analysis to show that resistant and susceptible genotypes cluster together when the dimensionality of hormone measurements are reduced. Why wasn't this done? The authors mention that "We understand and appreciate the suggestion but depending on the field relevance these data were needed to explain the results and hence were used". I agree that their method can be good for gaining the mechanistic insight, and their explanations make sense, however they cannot use that data to claim that resistant and susceptible genotypes can be distinguished based on hormonal profiles.
PCA was presented as an added tool to explain the results in a gist. We understand the redundancy but overall, the analysis is apt and appropriate.
5) Regarding SA-JA crosstalk. I know and agree that JA can be notoriously hard to measure. The authors should still provide references in the discussion section that show that SA and JA behave differently in the plums and as such this line of research warrants a separate study.
We have added a small section at the end to reflect that SA and JA is dealt separately
Reviewer 2 Report
I agree to publish in this form
Author Response
Thank you very much.
Round 3
Reviewer 1 Report
I don't disagree with the authors that this manuscript is valuable deserves publication, my concern was rather with proper choice and description of statistical tools used for establishing significance of the research. That said, I think the manuscript is acceptable for publication with two caveats:
1) This passage "For a particular compound the data were analyzed separately and p-value for all way interactions was provided. For that compound, origin, genotype, and tolerance were fixed factors. Except the p-values mentioned in the legend, interactions were statistically significant for remaining compounds." from the response should really be included either in the methods section or in the figure legend as it would prevent a lot of confusion. It is always a good practice to make it crystal clear what the p-value represents for any sort of statistical analysis.
2) Thus far I have seen only 4 panels in Fig. 4, all of which are labeled as "European". I haven't seen any lower panels labeled as "Japanese" that the authors mention in any of the versions of the manuscript. Is there something wrong with the PDF? I'd like the authors to carefully double check that the final PDF is rendered correctly and includes all figures.
Author Response
Dear Reviewer
Thanks a lot for your clarifications.
1) This passage "For a particular compound the data were analyzed separately and p-value for all way interactions was provided. For that compound, origin, genotype, and tolerance were fixed factors. Except the p-values mentioned in the legend, interactions were statistically significant for remaining compounds." from the response should really be included either in the methods section or in the figure legend as it would prevent a lot of confusion. It is always a good practice to make it crystal clear what the p-value represents for any sort of statistical analysis.
As you had suggested, which I fully agree on the clarity of the figures and now we have added that in the figure legend for figures 1 and 2.
2) Thus far I have seen only 4 panels in Fig. 4, all of which are labeled as "European". I haven't seen any lower panels labeled as "Japanese" that the authors mention in any of the versions of the manuscript. Is there something wrong with the PDF? I'd like the authors to carefully double check that the final PDF is rendered correctly and includes all figures.
Thanks again for this catch - We have no idea how or why half of the figure got cut off in the PDF. I could be lost in the copy and paste as evben fresh copy and paste was doing the same or the formatting was messed up; Nevertheless we saved the Figure in a different format as a different file and now it is clear and ou can see both EP and JP in the figure.
Best wishes
Jay